# Histomorphometric, Immunohistochemical, Ultrastructural Characterization of a Nano-Hydroxyapatite/Beta-Tricalcium Phosphate Composite and a Bone Xenograft in Sub-Critical Size Bone Defect in Rat Calvaria

**DOI:** 10.3390/ma13204598

**Published:** 2020-10-15

**Authors:** Igor da Silva Brum, Lucio Frigo, Renan Lana Devita, Jorge Luís da Silva Pires, Victor Hugo Vieira de Oliveira, Ana Lucia Rosa Nascimento, Jorge José de Carvalho

**Affiliations:** 1Implantology Department, State University of Rio de Janeiro, Rio de Janeiro 20550-900, Brazil; jorgepires45@gmail.com; 2Periodontology Department, Universidade Guarulhos, Guarulhos 07023-070, São Paulo, Brazil; luciofrigo@uol.com.br; 3Orthodontics Department, State University Barcelona, 08193 Barcelona, Spain; doctornan28@gmail.com; 4Biology Department, State University of Rio de Janeiro, Rio de Janeiro 20550-900, Brazil; victorhvoabio@gmail.com (V.H.V.d.O.); ana.nascimento@uerj.br (A.L.R.N.); jjcarv@gmail.com (J.J.d.C.)

**Keywords:** synthetic, xenogenous, nano-hydroxyapatite, beta-tricalcium phosphate

## Abstract

Nowadays, we can observe a worldwide trend towards the development of synthetic biomaterials. Several studies have been conducted to better understand the cellular mechanisms involved in the processes of inflammation and bone healing related to living tissues. The aim of this study was to evaluate tissue behaviors of two different types of biomaterials: synthetic nano-hydroxyapatite/beta-tricalcium phosphate composite and bone xenograft in sub-critical bone defects in rat calvaria. Twenty-four rats underwent experimental surgery in which two 3 mm defects in each cavity were tested. Rats were divided into two groups: Group 1 used xenogen hydroxyapatite (Bio Oss™); Group 2 used synthetic nano-hydroxyapatite/beta-tricalcium phosphate (Blue Bone™). Sixty days after surgery, calvaria bone defects were filled with biomaterial, animals were euthanized, and tissues were stained with Masson’s trichrome and periodic acid–Schiff (PAS) techniques, immune-labeled with anti-TNF-α and anti-MMP-9, and electron microscopy analyses were also performed. Histomorphometric analysis indicated a greater presence of protein matrix in Group 2, in addition to higher levels of TNF-α and MMP-9. Ultrastructural analysis showed that biomaterial fibroblasts were associated with the tissue regeneration stage. Paired statistical data indicated that Blue Bone™ can improve bone formation/remodeling when compared to biomaterials of xenogenous origin.

## 1. Introduction

Biomaterial evolution has provided countless discoveries in the bone-guided regeneration research field, leading to the development of bioactive materials. Nowadays, it is possible to employ direct matrix deposition and reabsorption modulating cells involved in various tissue regeneration processes [1]. Among these materials, bone xenograft has been extensively studied and has provided extensive literature with new scientific evidence [2].

Currently, there are numerous hydroxyapatite (HA) forms from different sources available on the market, and only a few biomaterials are composed exclusively of beta-tricalcium phosphate (β-TCP). Bio Oss™, the world’s leading hydroxyapatite product, from bovine origin (xenogen), is composed of 100% hydroxyapatite, and many studies support its osteoconductor capabilities [3,4,5,6].

However, as biotechnology has evolved, other materials have emerged and have different compositions and origins, with the aim to provide materials free from genetic products, the so-called synthetics (alloplastics). Song et al. concluded that a combination of synthetic HA and β-TCP in different proportions (ratios) could lead to rapid replacement of newly formed bone provided by β-TCP, with a slow absorption of hydroxyapatite, to maintain the volume of the grafted area [7].

In addition, nano-biotechnology has supported not only the chemical relationship between nano-HA and nano-β-TCP, but also the fact that the nanometric structure could help in the ossification process. The powders have specific physicochemical and biological properties, which—because of their nanometric dimensions, large surface area, great interfacial compatibility, and capacity to interact with the cells—promote adhesion, migration, proliferation, and cellular distinction [8,9]. Consequently, because of all these characteristics, other biomaterials have been developed by mixing nano-HA and nano (β-TCP) in different ratios, with the aim to provide advantages of nano-technology-processed biomaterials over conventional HA, whether synthetic or xenogenic, in relation to the processes of osteoconduction and osteoinduction [10].

There are essentially two ways to analyze bone morphology: (1) isotopic collagen analysis (slightly less destructive and occasionally less accurate) and (2) histological analysis (less costly and destructive) [11]. Therefore, histological analyses are indicated to determine and quantify innumerable biological characteristics, among them the formation of bone matrix, osteoblasts, osteocytes, osteoclasts, and pro-inflammatory factors, among others.

Through immunohistochemistry [12] it is possible to detect and quantify antigen–antibody signals and inflammatory mediators involved in the bone repair process. As an example, markers such as metalloproteinase (MMP)-9, tumor necrosis factor (TNF)-α, and periodic acid–Schiff (PAS) staining directly related to this process can be used.

There are currently 24 subtypes of metalloproteinases (MMPs) that have been described in humans, rodents, and amphibians [13]. MMP transcripts are commonly expressed at low levels, but these levels can increase rapidly when tissues undergo remodeling, such as with inflammation, and wound healing. Particularly, MMP-9 (also known as gelatinase B) is a multi-domain enzyme that works in acute and chronic diseases related to inflammatory and neoplastic responses. The main cells responsible for secreting MMP-9 are macrophages and neutrophils [14]. One of the main reasons for studying MMP-9 is because it is essential in initiating osteoclastic resorption, where it triggers the process of removing the collagen layer from the bone surface before demineralization can begin [15].

Pro-inflammatory cytokines, such as interleukin IL-2, IL-6, and tumor necrosis factor-α (TNF-α), are directly related to bone resorption through inflammatory processes. In particular, the expression of TNF-α in areas of bone defects can disturb and impair bone regeneration [16]. TNF-α is produced by macrophages and many other cells, including CD4 + lymphocytes, neutrophils, and mast cells [17].

Development of the PAS (periodic acid–Schiff) technique has been an important histological marker in the bone remodeling process because it is used to mark bone growth [18]. Osteoblast organelles, which contain bone matrix precursors, are stained pink by the PAS technique [19]. PAS staining is an important method for labeling proteoglycans and collagen in the formation of the extracellular matrix. Recent research has demonstrated the presence of several collagen types and proteoglycans that are distinctly expressed in cartilage, in the passage from cartilage to bone, and in extracellular bone matrixes. These discoveries indicate the complexity of the extracellular skeletal matrix as well as its fluid expression. While the composition of cartilage and extracellular bone matrixes is well-known, the function of each macromolecule that constitutes these matrixes and their developmental regulation is not clearly comprehended [20,21].

Scanning electron microscopy (SEM) is used in studying guided bone regeneration to identify and quantify cells or particles involved in the process. Regarding hydroxyapatite, it is possible to use SEM to measure its size and identify its composition and distribution through microanalysis [22].

The aim of the present study was to compare two biomaterials, nano-hydroxyapatite/beta-tricalcium phosphate composite and bone xenograft, in sub-critical defects in rat calvaria and conduct a histomorphometric analysis using Masson’s trichrome and periodic acid–Schiff staining, in addition to TNF-α and MMP-9-immunostaining, and scanning electron microscopy.

## 2. Materials and Methods

### 2.1. Animals

The experimental protocol was approved by the local Animal Ethics Committee (#001/2019). Forty-eight adult male *Wistar* rats were used, 200–220 g body weight, and they were provided by Roberto Alcantara of the Biology Institute of Rio de Janeiro State University. Animals were maintained in individual cages with *ad libitum* access to food and water. The light/dark cycle (lights on at 7:00 a.m., off at 7:00 p.m.) and temperature (22 °C) were kept constant. Rats were divided into two experimental groups (*n* = 24) after surgical procedures.

#### 2.1.1. Bone Defect Surgical Procedures

Rats were anesthetized with ketamine hydrochloride/xylazine solution (1/1, 0.1 mg/kg, i.p.). The dorsal cranium was trichotomized, and a sagittal incision was made using a sterile surgical scalp. The skin and periosteum were cleared in both parietal regions, and bone defects were created. Bone defects were achieved using a sterilized punch (cutting edge Ø 3 mm). Bone fragments were carefully removed to avoid damage to the dura mater and related blood vessels. After biomaterial insertion in bone defects, the animal skin was carefully placed and sutured with cotton wire.

#### 2.1.2. Group Distribution

Rats were distributed into two groups: Group 1 (*n* = 24) bone defects were filled with 0.1 g of bone xenograft (Bio Oss™, Geistlich, Switzerland); Group 2 (*n* = 24) bone defects were filled with 0.1 g of nano-hydroxyapatite/beta-tricalcium phosphate composite (80/20%) (Blue Bone™, Regener, Brazil).

#### 2.1.3. Euthanizing Procedure

Sixty days after biomaterial insertion, animals were euthanized with ketamine hydrochloride/xylazine solution (1/1, 0.3 mg/kg, i.p.) under anesthesia. Animals were decapitated, and the heads were used in the following histological procedures.

### 2.2. Morphological Analysis Protocol

Animal heads were trimmed and decalcified in EDTA (7.0%) in phosphate-buffered saline (PBS) (0.1 M, pH 7.4) for 40 days. Specimens were washed in distilled water, dehydrated in alcohol (70, 95, 100%), clarified in dimethylbenzene, and embedded in Paraplast™ (Sigma-Aldrich, St. Louis, MO, USA) at 65 °C. Serial sections of 7 µm were cut using a microtome (LEICA, Nussloch, Germany) and collected on silanized slides.

### 2.3. Masson’s Trichrome Staining Protocol

The slides were dewaxed and then rehydrated with alcohol (100, 95, 70%), rinsed in distilled water, and immersed in Weigert’s iron hematoxylin (Sigma-Aldrich, St. Louis, MO, USA) for 10 min. After rinsing in distilled water, the slides were immersed in Biebrich fuchsin solution (Sigma-Aldrich) for 15 min, rinsed again in distilled water, and differentiated in phospho-molybdovanic acid solution for 10 min. Slides were then submerged in aniline blue solution for 5 min, rinsed with distilled water, dehydrated, and coverslipped.

### 2.4. Immunohistochemistry Protocol

Slides were deparaffinized, rehydrated, and immersed in hydrogen peroxide (3%) for 15 min to neutralize endogenous peroxidase. Sections were then washed in phosphate-buffered saline (PBS) and submitted to antigen retrieval using a citrate buffer solution, pH 6, heated to 60 °C for 20 min. Unspecific antigen cross-reaction was ascertained using PBS containing (3%) albumin bovine serum for 20 min. Afterwards, slides were incubated with antibody anti-MMP-9 (1:200) (Santa Cruz Biotechnology Inc., Santa Cruz, CA, USA) and TNF-α (1:200) (Santa Cruz Biotechnology Inc., USA) in a humidified chamber at 4 °C overnight. Sections were incubated using the VECTASTAIN^®^ Universal Quick HRP Kit (Vector Labs., Burlingame, CA, USA) detection system, revealed with DAB (3,3-diaminobenzidine) (Sigma-Aldrich, St. Louis, MO, USA), and counterstained with hematoxylin (Sigma-Aldrich, St. Louis, MO, USA).

### 2.5. Periodic Acid–Schiff (PAS) Protocol

Slides were deparaffinized and rehydrated with alcohol (100, 95, 70%), washed in distilled water, immersed in periodic acid (1%) for 5 min, and rinsed again in distilled water. Slides were immersed in Schiff’s reagent (Fuchsin Basic (1%), Sodium metabisulphite (2%) in HCl (2%) solution) for 5–15 min, followed by a wash in running tap water for 5–10 min and counter-stained with Herri’s hematoxylin for 15 s.

### 2.6. Image Acquisition and Histomorphometry

Three randomized Masson’s trichrome, PAS-stained, and immune-labeled (TNF-α, MMP-9) slides were photographed with a photomicroscope (Carl Zeiss-JVC TK-1270 color video camera, Oberkochen, Germany) at 400 magnification. Images were quantified using GraphPad Prism Version 8.0 (GraphPad, San Diego, CA, USA).

Blue-stained areas in Masson’s trichrome and pink-stained areas in PAS were considered. Brownish staining was considered as immune-marked to anti-TNF-α and anti-MMP-9 (Figure 1).

### 2.7. Statistical Analysis

Data were analyzed using one-way ANOVA followed by a Wilcoxon Matched-Pairs test (*p* < 0.05). All analyses were conducted with specific software (GraphPad Prism Version 8.0 and BioEstat 5.0) (Figure 2).

### 2.8. Scanning Electron Microscopy Protocol

Immediately after euthanizing, animal heads were trimmed and surgical sites for biomaterial insertion were re-exposed and immersed in glutaraldehyde (2.%) in sodium cacodylate buffer (0.1 M), pH 7.4, temperature 48 °C, for 12 h. Post-fixation included osmium tetroxide (1.0%) and potassium ferrocyanide (0.8%) in cacodylate buffer (0.1 M) for 1 h in a dark room. Three changes of sodium cacodylate buffer (0.2 M) in distilled water (pH 7.4) for 1 h followed. After rinsing, specimens were dehydrated in sequential ethanol grades (25–100%). Specimens were immersed in hexamethyldisilazane (10 min) and placed inside an evaporation chamber for drying. Colloidal silver adhesive was used to mount specimens on aluminum stubs (Electron Microscopy Sciences, Hatfield, PA, USA). The critical point was achieved (critical point dryer—CPD 030, Bal-Tec, Oberkochen, Germany), and specimens were sputter-coated with gold (cool sputter coater dryer—SCD 005, Bal-Tec, Oberkochen, Germany).

### 2.9. Scanning Electron Microscopy Analysis

Coated surfaces were observed and described under a scanning electron microscope (Quanta 250 FEV, Thermo Fischer Scientific, Waltham, MA, USA) by two experienced, precalibrated, blinded evaluators. Magnifications included 5000× to evaluate biomaterial homogeneity, 15,000× to evaluate cell clusters and architecture, and 20,000× to evaluate specific cell types.

## 3. Results

### 3.1. General Observations

Animals did not show pain-related behavior or feeding/drinking disturbances. Skin wounds healed, and hair started to grow over healed wounds.

All 24 animals were sacrificed 60 days after the bone defect procedures. No signs of hemorrhage, edema, or infection were observed.

### 3.2. Masson’s Trichrome Histomorphometry

Masson’s trichrome showed bluish areas in the bone matrix, due to the chemical affinity of aniline blue to collagen fibrils, and suggested new bone formation, as observed in photomicrographs of the central region of newly formed bone. In Group 2 (nano-hydroxyapatite/beta-tricalcium phosphate composite), very limited or no evidence of the biomaterial was observed. A large area of blue staining was evident, and a rich vascular bed frequently full of blood cells was also observed. On the other hand, in Group 1 (bone xenograft), very few blue areas were observed, biomaterial was easily observed, and the vascular bed was fairly discrete (Figure 3 and Figure 4).

### 3.3. PAS Histomorphometry

The PAS method is based on Schiff’s periodic acid oxidizing hydroxyl and amino/alkyl amine chemical groups, forming a magenta colored complex. It detected polysaccharides, glycoproteins, and glycolipids, suggesting new bone formation. In Group 2 (nano-hydroxyapatite/beta-tricalcium phosphate composite), fairly intense magenta staining was observed. It was concentrated in new bone matrix borders. In Group 1 (bone xenograft), in addition to a light magenta staining, the tissue architecture differed. The higher cell concentration and close proximity suggested immature bone tissue formation (Figure 5 and Figure 6).

### 3.4. TNF-α Immunostaining

TNF-α was identified as light brown deposits inside the cells and bone matrix. In Group 2 (nano-hydroxyapatite/beta-tricalcium phosphate composite), a more intense immunostaining signal was observed in the bone matrix, mainly in locations more densely crowded with cells. In Group 1 (bone xenograft), the immunostaining signal was more disperse and less related to cell concentration (Figure 7 and Figure 8).

### 3.5. MMP-9 Immunostaining

MMP-9 was identified as light brown deposits inside the cells and bone matrix. In Group 2 (nano-hydroxyapatite/beta-tricalcium phosphate composite), the immunostaining signal was fairly intense and located in the extracellular matrix closer to cell crowds. Cell-free matrix was essentially devoid of a signal. In Group 1 (bone xenograft), an evident immunostaining signal was also observed; however, it was concentrated only in a few areas (Figure 9 and Figure 10).

### 3.6. Scanning Electron Microscopy

In the scanning electron photomicrographs, it was possible to identify fibroblasts after the osseointegration process. Fibroblasts displayed the classical flat and spindle shape, with cytoplasm projections following collagen bundles. In addition, fine collagen fibrils could be observed alone or in conjunction with other types of matrix proteins. They formed a reticular structure mainly on the matrix borders (Figure 11).

## 4. Discussion

Worldwide advances in the field of biomaterial research have propelled the great need to identify biomaterials that harbor specific properties for use in surgical interventions. Several authors have researched new alternatives that provide more favorable cellular responses [23,24].

In this respect, our study tested a new nano-HA/β-TCP composite (Blue Bone™) against a successfully used biomaterial, xenogenic HA (Bio Oss™), which has gathered an extensive number of studies over the years to support its use [3,4,5,6]. We found that some inflammatory mediators and enzymes related to bone matrix remodeling were more promptly expressed with the nano-HA/β-TCP composite, supporting histological findings that suggest an improvement in new bone matrix production, over xenogenic HA, in experimental conditions.

Calvarial bone defects have been used in the last 35 years as in vivo models to evaluate and compare bone replacement materials and regenerative materials [25].

A study on rat calvaria assessed the effectiveness of bone reconstruction by using a variety of biomaterials: bovine bone, refined hydroxyapatite (HA), demineralized bone matrix (DBM), and purified bone collagen (COLL). It was concluded that HA and DBM powder extracted from bovine bone tissue could also be used to repair bone defects and demonstrated enough potential to be used in clinical studies [26]. Results of the current study, similarly, indicated that the conjunction of different biomaterials (nano-HA with β-TCP) presented a more favorable result than using bone xenograft alone.

A comparative biomaterial study conducted by Anghelescu et al., albeit in rat tibial bone, demonstrated that xenogenic HA (Bio Oss™) and β-TCP exhibited higher platelet endothelial cell adhesion molecule (PECAM 1) and vascular endothelial growth factor (VEGF) immunolabeling than other types of bioactive glasses [27].

Vascularization remains one of the obstacles that needs to be overcome in the bone graft design research field [28]. In this respect, the initial nano-HA/β-TCP composite (Blue Bone™) characterization study conducted by da Silva Brum et al. showed a large presence of vascular channels, which indicated that its nanometric structure had a positive result on the creation of blood channels [29].

Masson’s trichrome staining was used in the herein study to quantify the organic matrix formed between the bone xenograft group and the nano-HA/β-TCP compound group. A statistical difference was found, signaling that this nanotechnology, together with the union of the two biomaterials, favors cell proliferation [30]. Another study using a similar approach (HE and Masson’s trichrome staining) in rat calvaria evaluated bone growth in pure hydroxyapatite (HA) and hydroxyapatite (HA) with the addition of wollastonite–hydroxyapatite grafts. The authors found the wollastonite–hydroxyapatite group had more favorable bone growth [31].

Pang et al. in an in vitro inflammation essay revealed that HA particles positively regulated the expression of cytokines IL-1β, TNF-α, IL-6, IL-10, IFN-γ, and IL-2 [32]. TNF-α upregulation was found in this study as well, corroborating Pang et al.’s findings, albeit using a different technique (immunohistochemistry) and extended the upregulation profile, including MMP-9. These results were also corroborated by another study evaluating the cellular response of macrophages and osteoclasts, stating that hydroxyapatite could regulate the secretion of TNF-α and IL-6 by macrophages, which would directly affect osteoclast activity in relation to bone resorption [33].

Schiff’s periodic acid (PAS) is commonly used in investigations related to bone tissue. In a study using pure hydroxyapatite and bioglass in rat calvaria and HE staining to evaluate bone regeneration, in addition to PAS staining to quantify the area of blood vessels, it was found that bioglass promoted a better formation of blood vessels than pure hydroxyapatite [34]. These findings are in accordance with the herein study, which demonstrated more intense PAS staining in the nano-HA/β-TCP composite group than the bone xenogograft group. These results suggest further research is needed on bioglass and nano-HA in order to better understand the cell activation between these materials.

Optimizing bone regeneration is a strategic approach, notably in implantology, to circumvent critical clinical situations such as atrophic alveolar ridges. Increased alveolar ridges and maxillary sinus lift elevation are common approaches used in these situations that are based on the bone-inducing biomaterial potential. Grassi et al. restored the atrophic maxillary crest of a 52-year-old woman using a custom-designed lyophilized bone obtained from the corpse of a donor’s tibial hemi-plateau. The authors demonstrated a reduced rate of bone reabsorption after installation of the graft and implant [35]. A sinus lift was conducted, also in human subject, using the nano-HA/β-TCP composite (Blue Bone™) and successfully achieved new bone formation with the installed implant [30]. Research on nanoscaffold use in alveolar bone regeneration is an active field. Different compositions and forms of nanomaterials have been tested, including nanoparticles, nanofibers, nanotubes, nanosheets, and nanospheres, and they are frequently associated with growth factors [36].

The main limitations of this study include an evaluation period of only sixty days and the limited profile of cell mediators considered.

## 5. Conclusions

-The nano-HA/β-TCP composite presented better conditions for bone matrix formation when compared to bone xenograft.-MMP-9 and TNF-α up-regulation suggested the cellular response to the bone remodeling process was more favorable in the Nano-HA/β-TCP composite group.-The PAS staining technique suggested the Nano-HA/β-TCP composite induced an increase in glycoproteins, polysaccharides, and glycolipids.-The Nano-HA/β-TCP composite is a promising biomaterial that can replace bone xenograft.

## Figures and Tables

**Figure 1 materials-13-04598-f001:**
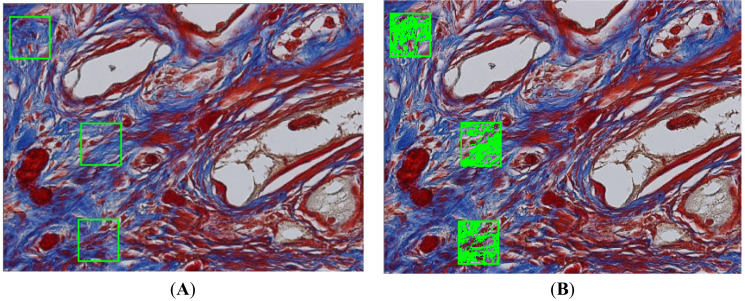
Histomorphometric analysis performed on a Masson’s trichrome stained slide. We can observe in the representative scheme performed by GraphPad Prism Version 8.0 (GraphPad, San Diego, CA, USA), the areas of delimitation chosen (**A**), to measure the percentage of blue coloration representing newly formed matrix (**B**).

**Figure 2 materials-13-04598-f002:**
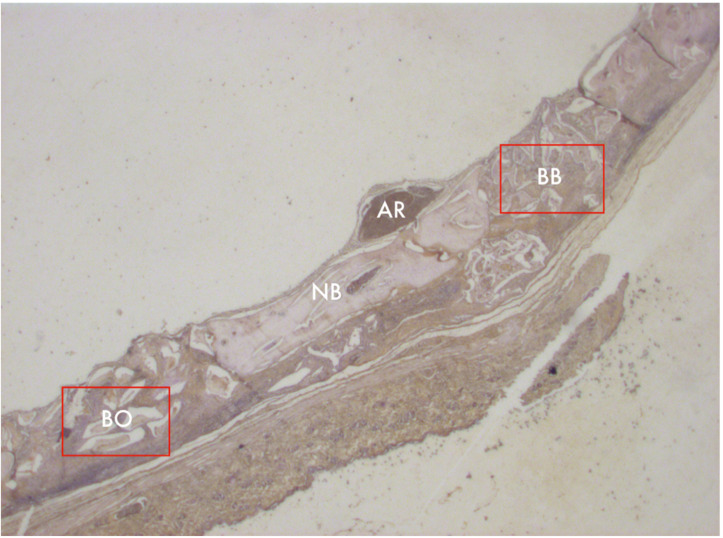
Photomicrograph of an animal calvaria sample 8 weeks after surgery; 1–25× magnification. BO = Bio Oss; BB = Blue Bone; NB = native bone; AR = artery.

**Figure 3 materials-13-04598-f003:**
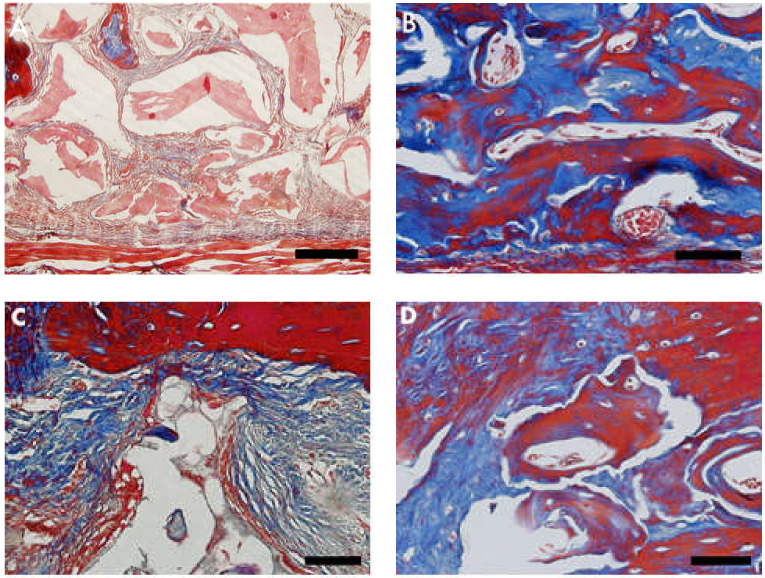
Photomicrographs of Masson’s trichrome stained slides. Bluish color indicates collagen fibrils in bone matrix. (**A,C**) Group 1 (bone xenograft), (**B,D**) Group 2 (nano-hydroxyapatite/beta-tricalcium phosphate composite ). Scale bar = 100 µm, 400× magnification.

**Figure 4 materials-13-04598-f004:**
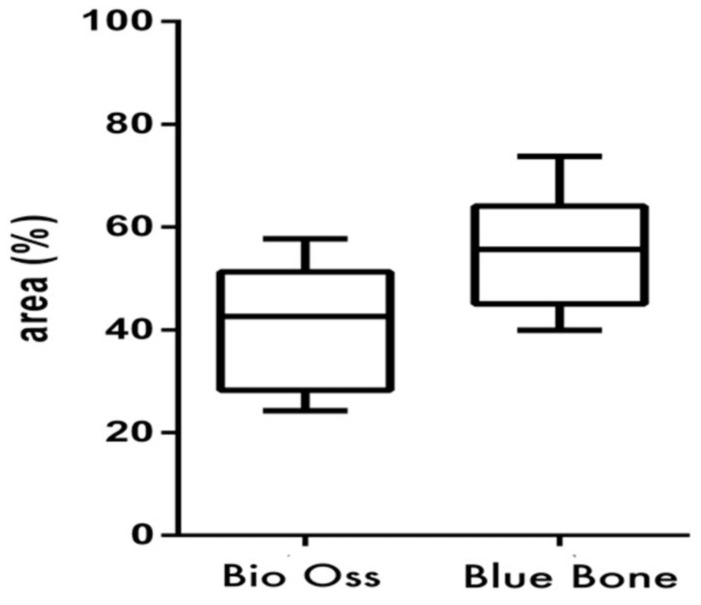
(Graphic 1) Histomorphometric analysis of Masson’s trichrome staining showed higher collagen content in Group 2 (nano-hydroxyapatite/beta-tricalcium phosphate composite) than in Group 1 (bone xenograft). A statistical difference was also found between Group 1 (bone xenograft) and Group 2 (nano-hydroxyapatite/beta-tricalcium phosphate composite), (*p* = 0.0313).

**Figure 5 materials-13-04598-f005:**
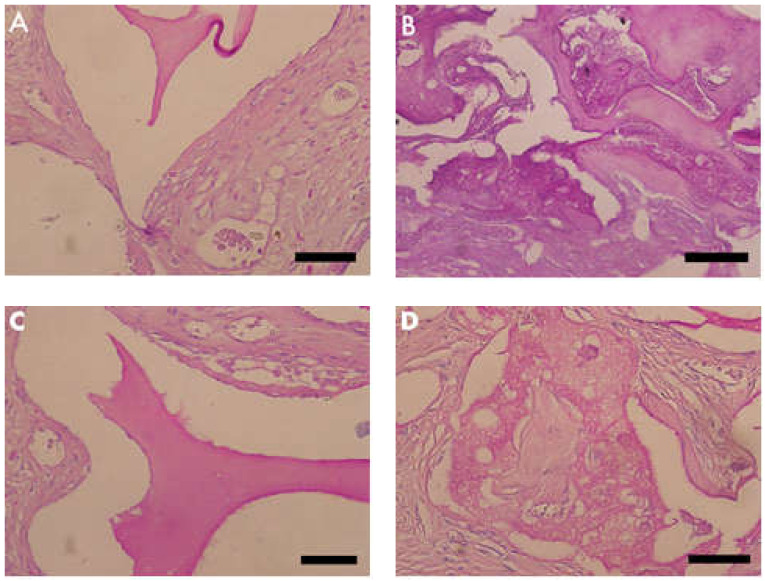
Photomicrographs of periodic acid–Schiff (PAS)-stained slides. The pink color indicates precursors of the bone matrix. (**A,C**) Group 1 (bone xenograft), (**B,D**) Group 2 (nano-hydroxyapatite/beta-tricalcium phosphate composite). Scale bar = 100 µm, 400× magnification.

**Figure 6 materials-13-04598-f006:**
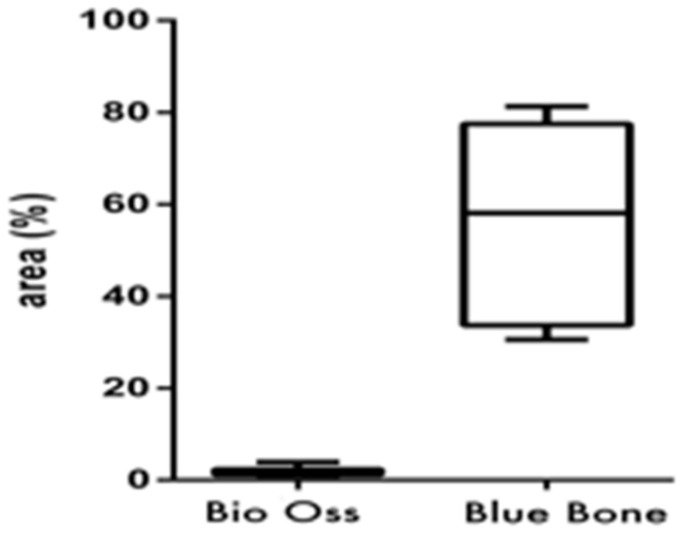
(Graphic 2) Histomorphometric analysis of PAS staining showed higher polysaccharides, glycoproteins, and glycolipid content in Group 2 (nano-hydroxyapatite/beta-tricalcium phosphate composite) than Group 1 (bone xenograft). A statistical difference was also found between Group 1 (bone xenograft) and Group 2 (nano-hydroxyapatite/beta-tricalcium phosphate composite), (*p* = 0.0313).

**Figure 7 materials-13-04598-f007:**
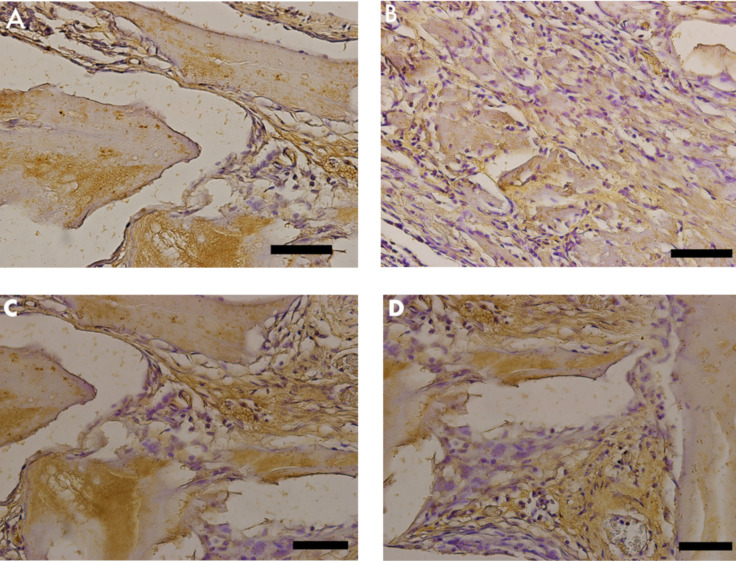
Photomicrographs of tumor necrosis factor (TNF)-α stained slides. Brown color indicates osteoclast activity. (**A,C**) Group 1 (bone xenograft), (**B,D**) Group 2 (nano-hydroxyapatite/beta-tricalcium phosphate composite). Scale bar = 100 µm, 400x magnification.

**Figure 8 materials-13-04598-f008:**
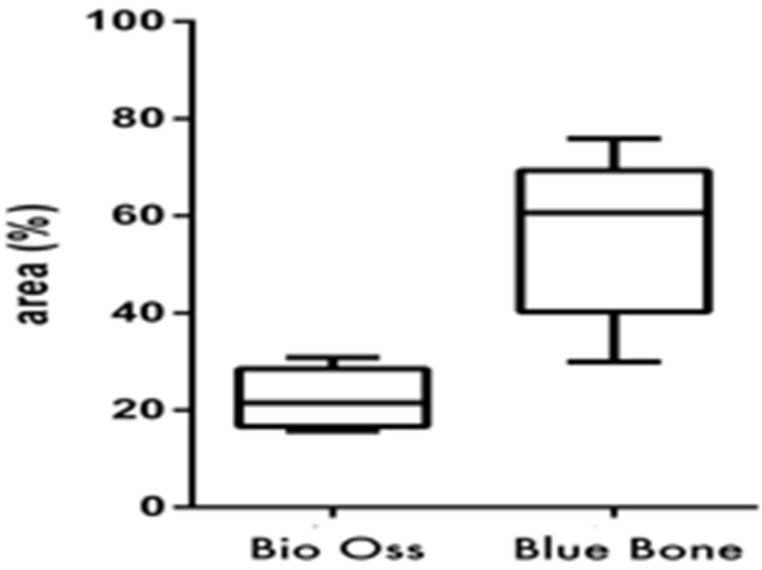
(Graphic 3) A statistical difference was also found between Group 1 (bone xenograft) and Group 2 (nano-hydroxyapatite/beta-tricalcium phosphate composite), signaling greater cellular activity in Group 2, (*p* = 0.0313).

**Figure 9 materials-13-04598-f009:**
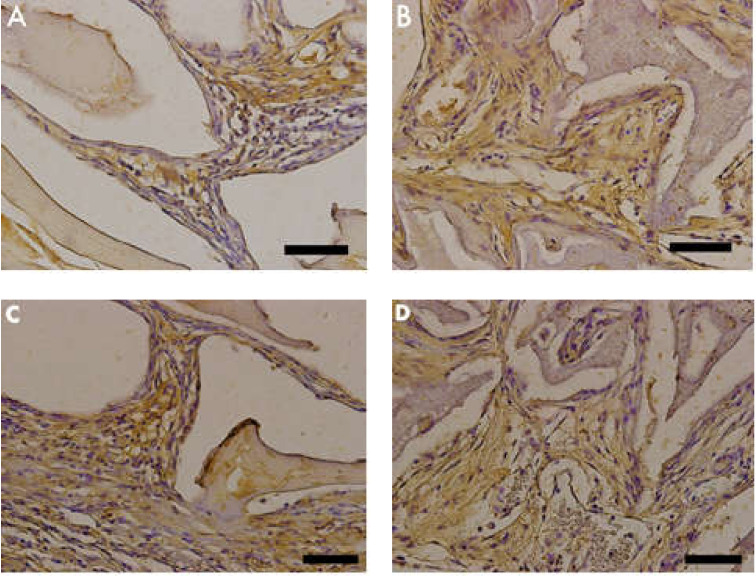
Photomicrographs of metalloproteinase (MMP)-9 stained slides. Brown color indicates osteoblastic activity. (**A,C**) Group 1 (bone xenograft), (**B,D**) Group 2 (nano-hydroxyapatite/beta-tricalcium phosphate composite). Scale bar = 100 µm, 400× magnification.

**Figure 10 materials-13-04598-f010:**
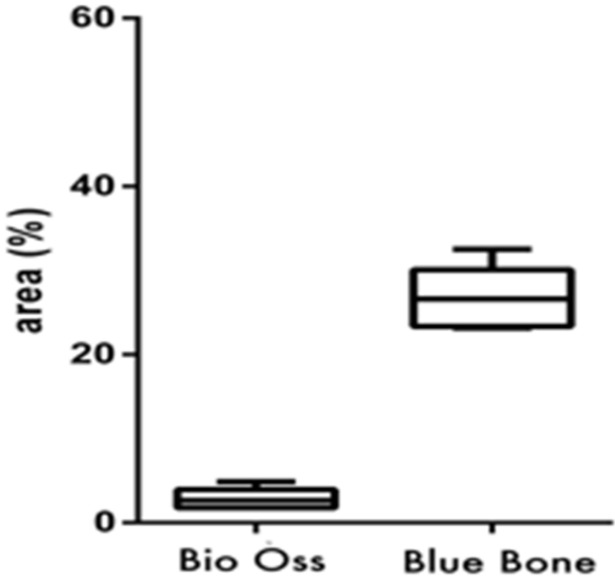
(Graphic 4) Histomorphometric analysis of MMP-9 staining showed that Group 2 (nano-hydroxyapatite/beta-tricalcium phosphate composite) had a larger immunostained area in comparison to that of Group 1 (bone xenograft), (*p* = 0.0313).

**Figure 11 materials-13-04598-f011:**
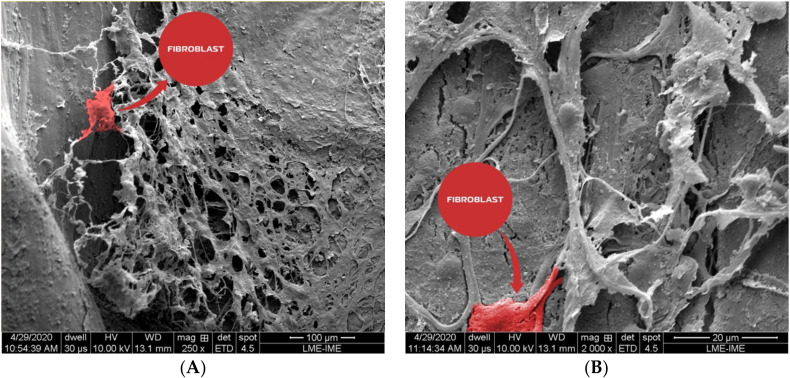
Photomicrographs showed the collagen fibrils that could be observed alone or in conjunction with other types of matrix proteins. They formed a reticular structure mainly on the matrix borders. An active fibroblast was observed in (**A**), (bone xenograft) and (**B**) (nano-hydroxyapatite/beta-tricalcium phosphate composite).

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
