# Peer review of "Histomorphometric, Immunohistochemical, Ultrastructural Characterization of a Nano-Hydroxyapatite/Beta-Tricalcium Phosphate Composite and a Bone Xenograft in Sub-Critical Size Bone Defect in Rat Calvaria"

_materials, 2020, doi:10.3390/ma13204598_

Round 1
Reviewer 1 Report
An interesting paper, Histomorphometric, Immunohistochemical,
Ultrastructural and Statistic Biomaterial Evaluation: Comparison Between Synthetic Nano-hydroxyapatite/Beta-tricalcium Phosphate and Hydroxyapatite in No Critic Defects in Rat Calvaria by da Silva Brum et. al.
Similar studies were performed, using other materials, with various composition, but based on the combination of (nano)HA and Tricalcium Phosphate, some using the same approach (damaged skull and injection).
The authors chose to test the product against a commercial product, that is all right, but why they selected Bio Oss™, and not other? At least in the introduction a few word justifying the selection would be required.
The Immunohistochemical section fails to explain the the sources of the primary antibodies. Please be specific on the sources.
The statistical section is a bit strange - since it declares only "one way Anova followed by Kruskal-Wallis analysis" Please explain the sliding form "One way Anova" to "Wilcoxon" and finally to "Spearman". If all these statistical analyses were applied, they should be declared as such in the statistical analysis.
The results section should require more than a picture. and a table and simply two lines of conclusions.
Moreover, the discussion section would also be more sound, discussing every finding of the authors by comparison to other such findings, if available), and to outline a bit the degree of novelty provided by the study.
Considering these, I think a major revision is required.
Author Response
v

Reviewer 2 Report
The authors were aimed to compare two biomaterials, nanohydroxyapatite / beta-tricalcium phosphate and hydroxyapatite in non-critical defects in rat calvaria in terms of histomorphometric analysis using, Masson's trichrome and Periodic Acid Schiff staining, in addition to TNF- α and MMP-9-immunostaining and scanning electron microscopy.
The structure of the manuscript appears adequate. The methodology is well described with enough experimental data and results to support the work.
Major issues: Please check typos and grammar thorough the text. Please remove redundant sentences within all the text.
Discussion section: Will be very useful for the readers, for the issue related to biomaterial research used for surgical interventions, as also stated from the l authors about alternatives that present more favorable cellular responses, to address also to the possibility of new therapeutical approaches i.e Freeze-dried Bone Custom-made Allografts (please see and discuss PMID: 32204393) as well the use of enginereed scaffolds.
Conclusion Section: This paragraph required a general revision to add some "take-home message".
Author Response
√

Round 2
Reviewer 1 Report
Overall, an interesting paper proposed by da Silva Brum et al, concerning the Histomorphometric, Immunohistochemical, Ultrastructural and Statistic Biomaterial Evaluation: Comparison Between Synthetic Nano-hydroxyapatite /Beta-tricalcium Phosphate and Hydroxyapatite in No Critic Defects in Rat Calvaria
The authors were producing modifications to the paper, which seems almost ready to publish. Provided that I thing some minor revisions of English language could be by the editorial team during paper processing, I suggest publishing.
Author Response
Dear..!!
The english edition was done at MDPI
thank you very much

Reviewer 2 Report
Authors have addressed to all comments. Thank you.
Author Response
Dear..!!
The english edition was done at mdpi
thank you very much
